# Damages to Himalayan White Pine (*Pinus wallichiana*) by Asiatic Black Bear (*Ursus thibetanus*) in Kaghan Valley, Pakistan

**Zaib Ullah** [1,*] **, Sajid Mahmood** [2] **, Zafar Iqbal** [3] **, Naveed Akhtar** [1] **, Muhammad Fiaz Khan** [2] **, Amir Said** [4] **, Mohammad Ayaz Khan** [5] **and Muhammad Arif** [6]

1   Department of Zoology, Hazara University Sub-Campus Battagram, Battagram 21040, Pakistan; akhtarzoologist@gmail.com
2   Department of Zoology, Hazara University Mansehra, Mansehra 21120, Pakistan; sajid_sbs12@yahoo.com (S.M.); fiazkhanhu333@gmail.com (M.F.K.)
3   Department of Botany, Hazara University Mansehra, Mansehra 21120, Pakistan; zafar.hu@yahoo.com
4   Department of Zoology, University of the Punjab, Lahore 54590, Pakistan; saidamir825@gmail.com
5   Sustainable Forest Management Project, Islamabad 44000, Pakistan; ayazkhan71@yahoo.com
6   Sustainable Forest Management Project, Peshawar 25130, Pakistan; orakzaipk@gmail.com
*   Correspondence: zaibullah_zoology@hu.edu.pk; Tel.: +92-334-934-9970

**Abstract:** Tree damage is one of the destructive behaviors of the Asiatic black bear (*Ursus thibetanus* G. (Baron) Cuvier, 1823), and this type of damage causes great economic loss to the forest. A survey about Himalayan white pine (*Pinus wallichiana* (A. B) Jacks, 1836) damages was conducted at Kaghan Valley, District Mansehra, Khyber Pakhtunkhwa, Pakistan. Field surveys were carried out within five major sites of Kaghan Valley, including Manshi reserve forest, Kamal Bann reserve forest, Malkandi reserve forest, Noori Bichla reserve forest, and some Guzara forests. Line transects and diameter at breast height (DBH) methods were selected for data collection. Eighteen transects were placed in different sites of the valley. A total of ($n = 201$) affected trees were observed from eighteen transects, along with a total population of 1081 trees with the encounter rate (ER: 0.657) and the mean DBH is $\bar{x} = 71.97$ cm. Among total damages, the most severe ($n = 39$: 19.4%) were fully damaged with a greater encounter rate. Bark stripping was made during the late winter season and used as foodstuff when natural food is limited in the area. In severe cases, the bear-stripped bark encircles from the entire tree trunk, which results in the drying of trees and, finally, falls. Among all five sites, Manshi reserve forest was greatly affected, where the highest number ($n = 76$) of tree damage, and ($n = 21$) the entire diameter of trunks were damaged. People of the study area claimed that the black bear causes great forest damage, as well as crop destruction that leads to high economic loss.

**Keywords:** Asiatic black bear; tree damages; Kaghan valley; *Pinus wallichiana*; transect; *ursus thibetanus*

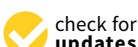



## 1. Introduction

Forest damages by the black bear were first time reported by Glover in 1955, damages of the redwood plant (*Sequoia sempervirens* (D. Don) Endl, 1847) in California; at that time his works were unpublished [1,2]. Bark removal from the trees in strips is a unique behavior of the black bear; strip marks vary in length from a few inches to several feet, while the width is three to twelve inches [3,4]. Mostly, bears remove bark from the base of the tree into an upward direction over the stem; in some cases bears stripped off the bark intact with the total width of the tree up to several feet from the ground [2,3]. Bears use their forefeet to begin the process of bark stripping, once removal of the bark was initiated they then unwrap the whole trunk of the tree [2,5]. Stripping of the bark was only recorded from the main trunk of the tree and not included the side branches [1,4]. These types of demarcation represent the territory of the individual bears, where they climb over the trees, during climbing they use incisor teeth to displace the cambial layer of the hard tree, and a major part of the unwrapped area was used for feeding purposes [1,2]. In some cases,

the total width of the tree has been removed during stripping and it is a major diet of the bear during late winter [1,5]. This process continues until the complete unwrapping and the trees are dry and weak. They no longer support their weight during high storms and finally fall to the ground [1,6].

The black bear emerges from the den during late winter (mid-January to mid-March) when the natural food source is limited in western Washington. At this time bears feed on false dandelion (*Hypochaeris radicata* (Carl Linnaeus), 1753), horsetail (*Equisetum arvense* (Carl Linnaeus), 1755), cow parsnip (*Heracleum lanatum* (W. Bartram), 1972), and skunk cabbage (*Lysichitum americanum* (Hultén & H.St.John), 1901) [5,7]. The bear starts subsequent foraging of bark peeling off the phloem tissues at the lower elevated area around April. Feeding upon vascular tissues can be extremely damaging or fatal to conifer trees [1,7]. Primarily, black bears target young conifer trees (15 to 25 years old) and up to 1000 trees per hectare. Incisor teeth play a vital role in phloem removal and leave clear tooth marks on the tree xylem [6–8]. These damages are very severe to the forest as bears always target the most energetic and healthy trees within the forest [6,9–14]. When the growth of the plant peaks during the May month, the phloem contains more sugars and bears continuously feed upon the sapwood of the coniferous trees [9,10]. Sugar (fructose, glucose, and sucrose) is used as the primary source of energy when bears emerge from the den during the late winter season [6,15]. A single black bear damaged 60–70 trees during foraging in the moist forest of western Washington [16]. Supplemental food sources and lures were used during the spring months to decrease forest (trees) damages. Once supplemental food is available to a black bear and later wild barriers mature and provide enough energy, it avoids tree damages throughout the season [1,3,10].

The present study was conducted in one of the important ecological zones of Pakistan, where a specific population of Asiatic black bears presents and causes tree damages in the forest. Here, in this case, bears mostly targeted Himalayan White Pine in Kaghan Valley. An intensive number of damages were observed during the field survey which badly affects the population of Himalayan White Pine constantly. These damages are unique to Asiatic black bears and mostly targeted specific trees when the food source is scarce. The greater economic loss was observed due to this behavior.

Objectives:

1. Investigation of tree damages (Himalayan White Pine) in Kaghan valley;
2. Point out highly damaged potential sites in Kaghan valley;
3. Investigate population abundance and shortage of food sources in the study area;
4. Investigate the age of the claw marks over trees.

## 2. Materials and Methods

### 2.1. Study Area

The present study was carried out in Kaghan Valley (22,000 ha) located north of District Mansehra, Khyber Pakhtunkhwa, Pakistan (Figure 1). The study area is surrounded by Azad Jammu and Kashmir (AJK), Battagram, Kohistan, and Northern areas of Gilgit-Baltistan [11].

Kaghan Valley of District Mansehra represents an important ecological zone in the province. The nearest town to Kaghan Valley is Balakot, while some other important villages of the valley include Paras, Kawai, Mahandri, Naran, and Kaghan. The valley possesses diverse topography, having mostly hilly areas and few planes. Rainfall occurs in the monsoon, summer, and winter seasons, the average rainfall recorded is 2500 mm per year. Snowfall starts at the end of October and continues till the end of February, and the snow may stay over mountains for several months. The winter is very severe, with heavy snowfall which is expected during late winter. The lowest minimum and highest maximum temperatures recorded are −6.0 °C in January and 30.5 °C in June, respectively [17].

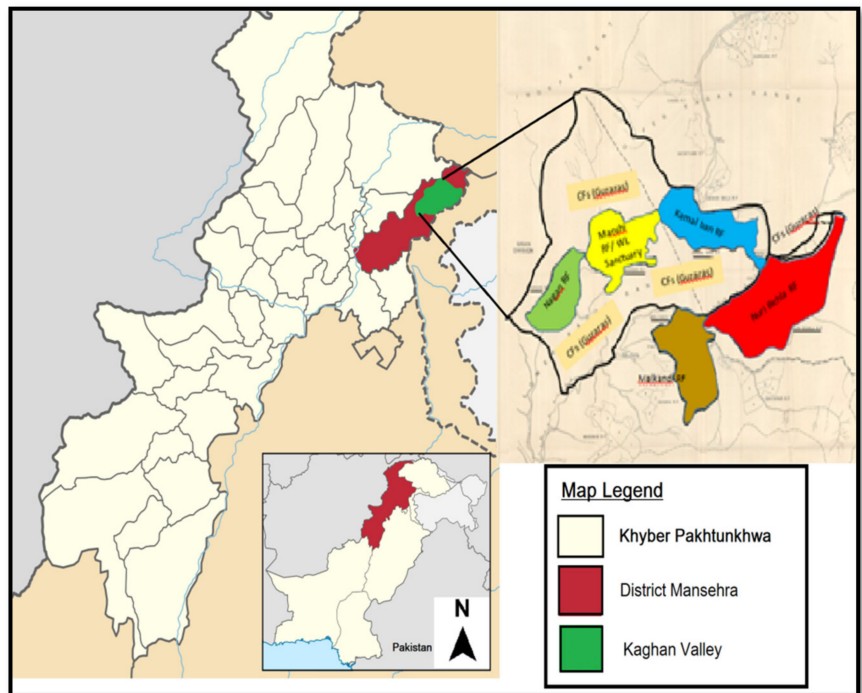

**Figure 1.** Map of Kaghan Valley, Pakistan.

The valley is surrounded by random parallel ranges and rises to 5291 m at Malika Parbat; from where the Kunhar River originated. The entire valley is about 96 km long and 24 km wide, covering a 945 square kilometer area. Land of the valley is used for different purposes containing 55% grazing, 24.6% forest, 2.6% agriculture, and the rest is built-up barren land or roads. Almost the entire valley is subject to the grazing of varying intensity and frequency [17].

The valley is subdivided into five major sites based on forest zones, including Manshi reserve forest (Manshi wildlife sanctuary: 2368 ha), Malkandi reserve forest (1923 ha), Kamal Bann reserve forest (2212 ha), Nuri Bichla reserve forest (1787 ha), Nagan reserve forest (1637 ha), and some Guzara forests (Bagheer: 2896 ha, Bhonja: 2208 ha, Ganila: 114 ha, and Hungrai: 415 ha) [11]. Coordinates of the study area range from (34°30.979′ N, 073°38.740′ E) to (34°43.24′ N, 073°30.822′ E); it is the moist temperate coniferous forest located on both sides of river Kunhar [11]. Elevation of the study area varies from 1200 m to 3500 m above sea level. The Asiatic black bear is present in every site of the study area but is mostly recorded in the Manshi reserve forest [11].

### 2.2. Methods

Data were collected during six months of field surveys from June to November 2019. Each survey was launched for fifteen days with help of the wildlife department of the District Mansehra. All the potential sites were searched for black bear claw marks (bark off) and recorded bear damages concerning coniferous trees (Himalayan white pine).

The line transect method was followed for counting the conifer trees and their damages; the length of transects was 300 m, while the width was 25 m. A total of eighteen transects were placed in different potential sites of the valley, the highest number of transects (four) in Manshi reserve forest, followed by Malakandi and Kamal Bann reserve forest (three apiece), Nuri Bichla, and Nagan reserve forest (two apiece), and four in Guzara forests (Bagheer, Bhonja, Ganila, and Hungrai: one in each). In this method, a team of three members was involved, two of them walked in a zigzag direction to search for bear damages, and one member of the team walked on the midline to record the data. Observations of the damaged conifer were recorded with the help of GPS coordinates (GARMIN etrex 30x FCC ID: IPH-01842, Taipei, Taiwan). Picture of damages was made

from every site of the forest with the help of a DSLR camera (Canon Eos 6d WG DS126401, Tokyo, Japan) along with two different lenses (Canon zoom lens EF 24–105 mm 1:4 L IS USM, Tokyo, Japan) and (Canon zoom lens EF 70–200 mm 1:2:8 L IS II USM, Tokyo, Japan). Claw marks and tree damages were also observed from a far distance with the help of binoculars (Nikon Aculon A211 12 × 50 No.6055245, Shanghai, China).

Damages of conifer trees were calculated with the help of the encounter rate (ER: number of sign/transect length). The encounter rate of damages was calculated for each site of the forest. The length and width of each damaged tree were calculated among all transects; for this purpose, we followed the Diameter at breast hight (DBH) method for trunk measurement of the damaged trees. Very high levels of conifer damages were recorded from Kaghan Valley; black bears targeted the blue pine (*Pinus wallichiana*) and used its bark as food during the winter season. These remarkable damages were observed from 0.457 m above the ground level; the length and width of the damaged trees vary from place to place within the forest and were recorded with the help of a measuring tape (10 m) scale during the field survey. The number of the conifer plants concerning their damages was noted from each transect, only mature trees (>10 years old) damages were examined and considered the affected trees of the study area.

The age of tree marks was also calculated from the damaged trees as once the damages are executed by a black bear, these marks persist for many years over the tree trunk. Bears leave these marks while climbing over trees, reproductive foraging, and sourcing food. These marks were made during the late winter season when natural food is scarce. Age of damages was differentiated from the degree of regrowth of stem bark; the growth rate was less in old trees as compared to fresh ones. Damages were divided into four categories based on their age; fresh (2–12 months), recent (1–2 years), old (2–4 years), and much older (more than four years).

## 3. Results

A total of eighteen lines-transects were randomly examined and a very high number of tree damages were recorded (Table 1). A total of ($n$ = 201) bark strip signs of a black bear were recorded; among these, the highest number of damages 38% ($n$ = 76) were found in Manshi reserve forest, followed by 20% ($n$ = 41) in Kamal Bann reserve forest, and 13% ($n$ = 26) in Malakandi reserve forest. The highest encounter rate was recorded from Manshi reserve forest (0.063), followed by Kamal Bann reserved forest, Ganila, and Bagheer guzara forests (0.045, 0.043, and 0.033, respectively). Black bears damage the trunk of the tree up to a specific distance from the ground, ranging from 0.274 to 0.701 m. damaged areas over trees varied from place to place and transect to transect. Based on the average, there was a 0.457 m area in length throughout the study.

**Table 1.** Detail of the transects in Kaghan Valley, Pakistan.

| Potential Sites | Manshi Reserve Forest | | | | Malakandi Reserve Forest | | | Kamal Bann Reserve Forest | | | Nuri Bichla Reserve Forest | | Nagan Reserve Forest | | Bagheer (GF) | Bhonja (GF) | Ganila (GF) | Hungrai (GF) | Total |
|---|---|---|---|---|---|---|---|---|---|---|---|---|---|---|---|---|---|---|---|
| Transects Compartment | C10 | C12 | C8 | C1 | C1 | C6 | C4 | C8 | C10 | C13 | C26 | C27 | C14 | C8 | C11 | C7 | C1 | C14 | 18 |
| Area (ha) | 115.3 | 245.2 | 170.4 | 105.9 | 99.14 | 90.2 | 105.2 | 77.6 | 141.6 | 90.64 | 125.4 | 145.2 | 113.3 | 140.0 | 233.099 | 158.00 | 103.6 | 163.08 | 2422.86 |
| No of Damaged Trees | 38 | 12 | 8 | 18 | 8 | 11 | 7 | 10 | 26 | 5 | 6 | 4 | 8 | 3 | 10 | 5 | 13 | 9 | 201 |
| Total Trees | 75 | 55 | 49 | 59 | 38 | 55 | 61 | 68 | 72 | 51 | 69 | 70 | 66 | 59 | 46 | 37 | 58 | 93 | 1081 |
| Encounter Rate (ER) | 0.126 | 0.04 | 0.026 | 0.06 | 0.026 | 0.03 | 0.023 | 0.033 | 0.086 | 0.016 | 0.02 | 0.013 | 0.026 | 0.01 | 0.033 | 0.016 | 0.043 | 0.03 | 0.657 |

C(Compartment), GF (Guzara Forest).

Bear damages are very severe to the conifer population, especially to the Himalayan white pine (*Pinus wallichiana*) in this area (Figure 2). Diameter at breast height (DBH) was measured at about 1.31064 m above the ground level, as it is considered a standard, also used by [1]. The average length ($\bar{x}$:55.53 cm) and width ($\bar{x}$:35.67 cm) of the scars was calculated from all 201 marks. Diameter at breast height was calculated for each mark, cumulatively ($\bar{x}$:71.97 cm) damaged area were recorded from all the trees. Comparison

between the DBH and stripped damages was carried out among all transects, and a ratio ($\bar{x}$:2.01) was calculated between these two parameters. Only (*n* = 37) fully damaged trunks were found; among these the highest number of damages were recorded at Manshi reserve forest (21), followed by Kamal Bann reserve forest (13), and Ganila guzara forest (3) (Table 2).

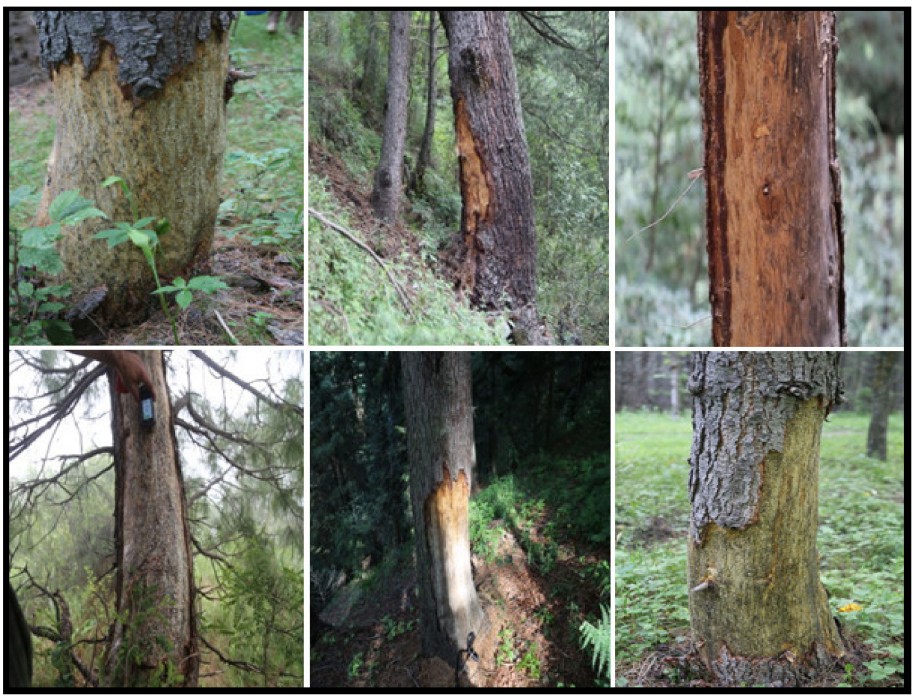

**Figure 2.** Photos of *Pinus wallichiana* damages in Kaghan Valley, Pakistan.

**Table 2.** Summary of the Himalayan white pine (*Pinus wallichiana*) damages with help of transects and DBH methods.

| Forest Zone | Area (ha) | Number of Transects | Total Length of Transects(m) | Total No of Conifers | Total Damages | Encounter Rate (ER) | Percentage (%) | Average Height from the Ground (m) | Damages Length (cm) $\bar{x}$ | Damage Width (cm) $\bar{x}$ |
|---|---|---|---|---|---|---|---|---|---|---|
| Manshi RF | 2368 | 4 | 1200 | 238 | 76 | 0.252 | 37.81 | 0.396 | 85.3 | 39.2 |
| Malakandi RF | 1923 | 3 | 900 | 154 | 26 | 0.079 | 12.93 | 0.304 | 48.7 | 32.8 |
| Kamal Bann RF | 2212 | 3 | 900 | 191 | 41 | 0.135 | 20.39 | 0.304 | 67.0 | 42.5 |
| Nuri Bichla RF | 1787 | 2 | 600 | 139 | 10 | 0.033 | 4.97 | 0.457 | 57.9 | 33.1 |
| Nagan RF | 1637 | 2 | 600 | 125 | 11 | 0.036 | 5.47 | 0.274 | 39.6 | 29.9 |
| Bagheer GF | 2896 | 1 | 300 | 46 | 10 | 0.033 | 4.97 | 0.609 | 54.8 | 44.6 |
| Bhonja GF | 2208 | 1 | 300 | 37 | 5 | 0.016 | 2.48 | 0.548 | 33.5 | 25.7 |
| Ganila GF | 114 | 1 | 300 | 58 | 13 | 0.043 | 6.46 | 0.701 | 82.2 | 52.6 |
| Hungrai GF | 415 | 1 | 300 | 93 | 9 | 0.03 | 4.47 | 0.518 | 30.4 | 20.7 |
| Total | 22,000 | 18 | 5400 | 1081 | 201 | 0.657 | 100 | 0.457 | 55.5 | 35.6 |

RF (reserve forest), GF (Guzara forest), DBH (diameter at breast height), and $\bar{x}$ (mean).

### *Age of Claw Marks on Trees*

The age of the claw marks was calculated for each sign, this was done with help of a field expert who calculated regrowth, length, and width of claw marks over trees. Among four categories, Old marks were observed with the highest number (*n* = 85: 42.28%), followed by recent (61: 30.34%), much older (39: 19.4%), and fresh marks (16: 7.96%).

## 4. Discussion

Some species of bear cause very severe damage to trees that persists over the tree trunk for a very long time. The diameter at breast height (DBH) method was used for damages of Japanese cypress (*Chamaecyparis obtusa*) by Japanese black bear (*Ursus thibetanus japonicus*). In this case, trees with a larger diameter were more affected than the smaller diameter trees, and nutritional components such as sugar, especially sucrose, were consumed by

Japanese black bears. A very little variation of sugar concentration of vascular tissues was recorded which was not correlated with stand age and DBH. Mass of the vascular tissue was positively correlated with DBH, but not with the standard age [2,18].

Among the conifer species, one of the most threatened species is the western redcedar, damaged by black bears (*Ursus americanus*) in the Pacific Northwest. Two methods were used during the collection of the data from 122 vulnerable stands into four different damaged categories, i.e., Total Loss; Salvage; Root Disease, and Combined Damage. The economic loss of the trees was ≤0.35% of the total present value under all types of damages while processing damaged trees (Salvage) was the most effective option. At the landscape scale, the worst-case situation (Total Loss) resulted in an estimated economic loss of $56/ha to bear damage. Most of the damages (92%) were observed from the ground survey, where the sign persists for more than two years and existed at a low-frequency level (1.5 bear damaged trees/ha) across the landscape. This represented that these damages were not uniformly distributed and that apparent impact varies with spatial scale [1,19].

Stripping of conifer bark behaviors by the Asiatic black bear was recorded from most of the areas from Japan. During this study, it is concluded that black bears mostly attacked conifer trees from early May to late July; during this period, the sap which can flow in the cambium layer is very high. On the other hand, bark stripping behavior was not found in Japanese bird cherry. As the result of this study, they concluded that bark stripping behavior was caused due to shortage of food sources, and maybe varied among different regions [20,21].

Our study revealed that Asiatic black bears cause severe damages to Himalayan white pine (*Pinus wallichiana*) which are consumed as foodstuff during the harsh winter season which is highly similar to the previous studies. The above-mentioned studies recommended that bear damages were not uniformly distributed. Similarly, in this case, these damages are restricted to a specific region that is Kaghan Valley (most of them are in Manshi reserve forest), as here the population of the Asiatic black bear is quite high and natural food scares during the harsh winter season. Black bears not only remove bark from a specific region; in most cases, they remove the entire surface of the tree trunk, and the tree is unable to transport water and nutrients from the ground to the upper parts. The fully damaged trunk is very dangerous for the forest population as it loses the transportation system of the trees and finally falls during high winds and storms.

## 5. Conclusions

From this study, it is concluded that severe damages have been caused by an Asiatic black bear (*Ursus thibetanus*) to Himalayan white pine (*Pinus wallichiana*) in the Kaghan Valley. A total of ($n$ = 201) affected trees of *P. wallichiana* have been observed; among these, the most severe ($n$ = 39: 19.4%) were fully damaged. Within the study area, the Manshi reserve forest was considered the most damaged spot as compared to other areas. Black bears targeted *P. wallichiana* and used its bark as a nutritional food during the winter season. Due to severe damages, the survival rate of such trees decreases as they slowly dried and, at last, they fell to the ground during a high storm. It is observed that such a destructive behavior of *U. thibetanus* results in the decline of *P. wallichiana* in Kaghan Valley Pakistan.

**Limitations and Future Research Directions**

This research indicated insights into potential damages of Himalayan white pine (*Pinus wallichiana)* by Asiatic black bears in Kaghan valley Pakistan. Black bears caused severe damages to conifer trees in the last few years, due to a shortage of food sources. These damages could be considered higher risks for the white pine population. Among the entire study area, Manshi reserve forest was greatly affected by black bear damages, and most of the trees become dried and fallen to the ground. However, this study has some limitations; one of the main limitations is that the study is only conducted in a limited area (Kaghan Valley). It would be more authentic if it covered the entire province or country. Secondly, the present study only represents a single species (white pine); more

concisely, the study would be more accurate and meaningful if all the species of the conifers were assessed.

Detailed surveys and in-depth studies are recommended for future research. This research provides the base for future studies that could cover all the conifer species damaged by various animals. We recommended that supplemental foodstuff for a black bear is required to minimize forest damages and economic loss.

**Ethical Approval Certificate**

It is stated that the Institutional Ethical Committee of Hazara University, Mansehra has thoroughly studied the objectives of this paper by Mr. Zaib Ullah and his co-researchers for Ethical Approval. The Committee recommends that there is no need for ethical approval, as this study does not have any ethical issues. The Committee did not find any issue whatsoever that can raise ethical concerns of the community in the study area.

**Author Contributions:** Z.U. conceived of the presented idea and wrote the manuscript. Z.U. designed the research methodology and performed the field survey in a meaningful way. S.M. and Z.I. verified the methods, encouraged Z.U. to investigate the current research gap, and supervised the findings of this work. N.A., M.F.K., A.S., M.A.K. and M.A. discussed the results and contributed to the final manuscript. Z.U., S.M. and N.A. reviewed the manuscript. All authors have read and agreed to the published version of the manuscript.

**Funding:** This research received no external funding.

**Acknowledgments:** I would like to thank Sajid Mahmood and Zafar Iqbal for their great help during the conduction of the research. I am also thankful to the Wildlife Department and Sustainable Forest Management project Khyber Pakhtunkhwa. My thanks extend to anyone who helped me in this work.

**Conflicts of Interest:** The authors declare no conflict of interest.

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
