# Peer review of "Damages to Himalayan White Pine (Pinus wallichiana) by Asiatic Black Bear (Ursus thibetanus) in Kaghan Valley, Pakistan"

_forests, doi:10.3390/f12081130_

Round 1

Reviewer 1 Report

The study investigated the tree damages which Asiatic black bears stripped bark by conducting field survey. The authors conducted line transect survey to reveal damage of conifer tree and age of clow marks. However, these results did not discuss in the manuscript. In addition, I could not find any sentences referring to the importance and novelty of this study, which makes it difficult to evaluate this study. The following are my comments.

  1. There were no motivation and no objective in the Introduction. Why did the authors study tree damages by bears in your study area? What are your research objectives or aims? Please revise the Introduction thoroughly.
  2. Although there are many previous studies about bark stripping by bears, including Asiatic black bears, the authors did not have sufficient discussion and used some references which did not support their description.
  3. Many descriptions in each section were inappropriate and inadequate in Methods and Results.

For example,

  • Some sentences in Results (lines 121-123, 146-147, 151-155, …etc.) should be in the Methods.
  • All tables have not been cited in the manuscript.
  • lines 125-126: What kind of tool did you use to measure the distance from the ground? The meters with five decimal places may not needed.

I think that “Front Matter” and “Research Manuscript Section” of Instructions for Authors (https://www.mdpi.com/journal/forests/instructions#manuscript) would help the authors to revise each section in the manuscript.

  1. The authors said that bark stripping was made during the late winter season and used as foodstuff when natural food is limited in the area in Abstract, Results, and Discussion. However, they did not investigate the season of tree damage and bear diet, and there were no references in these sentences.
  2. How did you determine the age of tree marks? Did you count the number of annual growth rings? Please explain it in detail in Method.
  3. The results from your survey (e.g., encounter rate, age of claw marks, size of bark stripping etc.) did not described in Discussion. Please reconsider your findings and rewrite Discussion thoroughly.
  4. The poaching reduced the most local population across Asia, including this study area, and Asiatic black bear is listed as vulnerable on the Red List by the IUCN. I strongly recommended to include conservation implications in the manuscript.
  5. While I am not a native English speaker, I have noticed that the manuscript requires a thorough review of English grammar and style.
  6. Please use italics for scientific names through the manuscript.

Author Response

  1. Extensive editing of English language and style required
  2. Response: All authors thoroughly read the manuscript and removed all grammatical and technical mistakes.
  1. Does the introduction provide sufficient background and include all relevant references?
  2. Response: introduction of the manuscript is reorganized and added some necessary paragraphs and cross-checked for references. Now the manuscript introduction is precise and meaningful
  1. There were no motivation and no objective in the Introduction. Why did the authors study tree damages by bears in your study area? What are your research objectives or aims? Please revise the Introduction thoroughly.
  2. Response:

-introduction of the manuscript was revised thoroughly by all authors,

-authors studied the economic loss of the tree damages that were caused by an Asiatic black bear because due to the current pace of damages, the Himalayan White Pine species decrease in the study area. The growth of this plant is quite slow, young trees were mostly targeted and used as a food source.

-objective of the study has been provided at the end of the introduction section.

  1. Although there are many previous studies about bark stripping by bears, including Asiatic black bears, the authors did not have sufficient discussion and used some references which did not support their description.
  2. Response: authors included a proper section in the discussion about the bark stripping of the Asiatic black bear. Proper references and citations have been provided accordingly.
  1. Many descriptions in each section were inappropriate and inadequate in Methods and Results.
  2. Response: all the relevant sections were placed according to the recommendation of the reviewer.
  3. Some sentences in Results (lines 121-123, 146-147, 151-155, …etc.) should be in the Methods.

All tables have not been cited in the manuscript.

Response: The entire sentences were replaced into their proper position as recommended by the reviewer.

  1. lines 125-126: What kind of tool did you use to measure the distance from the ground? The meters with five decimal places may not needed.
  2. Response: we used a meter for the measurement of height from ground level. Five decimals have been rounded into three decimals.
  1. I think that “Front Matter” and “Research Manuscript Section” of Instructions for Authors (https://www.mdpi.com/journal/forests/instructions#manuscript) would help the authors to revise each section in the manuscript.
  2. Response: each section of the manuscript was revised based on the above mention site of instructions.
  1. The authors said that bark stripping was made during the late winter season and used as foodstuff when natural food is limited in the area in Abstract, Results, and Discussion. However, they did not investigate the season of tree damage and bear diet, and there were no references in these sentences.
  2. Response: Based on claw marks and teeth stripping, the researcher investigated that bears used such barks as food stuff, this parameter was also confirmed by the wildlife and forest staff of the local area who practically observed such behaviors in the late winter season. Information’s from wildlife staff was also confirmed by researchers when they practically visited the study area for data collection during the late winter season.
  3. How did you determine the age of tree marks? Did you count the number of annual growth rings? Please explain it in detail in Method.
  4. Response: age of claw marks was calculated based on the nature of claw marks concerning the time, the entire damages were generally divided into different broad categories for easy assistance and data collection, data collection about the age of claw marks was also supported by field experts who already worked on this aspects.
  5. Please use italics for scientific names throughout the manuscript.
  1. Response: all the scientific names are now in italics throughout the manuscript.
  1. While I am not a native English speaker, I have noticed that the manuscript requires a thorough review of English grammar and style.
  1. Response: the entire manuscript has been reviewed by all the authors and removed all the grammatical mistakes.
  1. The poaching reduced the most local population across Asia, including this study area, and the Asiatic black bear is listed as vulnerable on the Red List by the IUCN. I strongly recommended to include conservation implications in the manuscript.
  1. Response: Conservation implications are added in this manuscript.
  2. The results from your survey (e.g., encounter rate, age of claw marks, size of bark stripping, etc.) did not describe in Discussion. Please reconsider your findings and rewrite Discussion thoroughly.
  3. Response: this section has been updated according to the recommendations of the reviewer. Now it is clear and meaningful.

Reviewer 2 Report

There are some minor grammatical and technical mistakes in the manuscript, please read the MS in depth for solving these minor mistakes, however the originality and novelty of the MS is suitable for the publication in forests.

Please check the MS for style and formatting according to the journal.

Author Response

  1. There are some minor grammatical and technical mistakes in the manuscript, please read the MS in depth for solving these minor mistakes; however the originality and novelty of the MS is suitable for the publication in forests.

Response:

-All the authors read and approved the final version of the manuscript that is free from any grammatical and technical mistakes. All authors read the manuscript line by line to avoid such minor mistakes in the final version. Furthermore, authors also added some necessary paragraph and shuffled the sentences to proper positions where it was best suited.

  1. Please check the MS for style and formatting according to the journal.

Response:

Instructions for the author were read again, for manuscript style and formatting, the entire manuscript was prepared according to the journal style.

Reviewer 3 Report

Review of Damages to Himalayan White Pine (Pinus wallichiana) by Asiatic Black Bear (Ursus thibetanus) in Kaghan Valley, Pakistan by Ullah et al.

This manuscript contains useful interesting data on bear damage rates at multiple sites in the Kaghan Valley of Pakistan. The Introduction would benefit from addition of an objectives statement at the end. The Methods section could be enhanced by adding description of the climate and forest composition and structure, and scientific names provided for all species common names only at the first instance. Also, the Methods section needs to clearly explain if this study focused on damage to white pine (L90) or blue pine (L109) or counted damage on all species (L113). I think the Discussion could be enhanced by comparing these Kaghan Valley results to other studies, including discussing major differences in tree size or age among this and other studies that might explain differences in damage (currently the Discussion is separated into two paragraphs describing other studies then one paragraph describing the Kaghan Valley results without detailed comparison to other studies. Also, I recommend adding discussion of the limitations of the study and recommendations for future research, and possibly implications for management/conservation/restoration. Finally, the Introduction and Discussion would be more impactful if they began with ‘big picture’ major problem statement (intro) and major finding (discussion). The authors appear to have good understanding and use of the English language but I recommend editing by someone who regularly writes in English for clarity. Below are some specific comments:

Usually scientific names are italicized.

L60 add citations for black bears damaging faster-growing trees e.g.:

Dagley, C.M.; Berrill, J-P.; Leonard, L.P.; Kim, Y.G. 2018. Restoration thinning enhances growth and diversity in mixed redwood/Douglas-fir stands in northern California, USA. Restoration Ecology 26(6): 1170-1179.

Perry, D.W.; Breshears, L.W.; Gradillas, G.E.; Berrill, J-P. 2016. Thinning Intensity and Ease-of-Access Increase Probability of Bear Damage in a Young Coast Redwood Forest. Journal of Biodiversity Management and Forestry 5(3):1-7.

L64 citation error – Nishi et al. should be numbered?  I believe you mean “severe”, but suggest rewording to “A single black bear damaged 60-70 trees during foraging in [forest type at location]”

L65 check citation numbering as I do not think [10] Berrill et al. reported number of trees damaged by a single bear.

L68 add objectives statement with research questions and/or hypotheses.

L78 Consider removing “GPS….” and only report the coordinates (not how they were collected).

L105 how was age of claw marks assessed? (detail please)

L107-108 this sentence not a method.

L112 I do not understand “measurable scale” - please clarify

L117 how were the transects located – random start point and azimuth? Systematic?

L121 can ER also be reported as percent damaged? (i.e., were undamaged trees also tallied?) Reporting percent of all trees damaged is more intuitive/interpretable than number/transect length.

L127 long sentence, and unclear if 0.4572 is a scar length or area or ?

L128 “…from the ground level…” is more of a method than a result.

L129-103 is blue pine same species as white pine? I think this will be clearer to the reader once the Methods section is clarified per previous comment.

L134 please specify that the length and width refer to the scar (not the tree itself)

L130-138 Some of this is Methods mixed in with Results – please move Methods to the Methods section

L140-142 Not a Result – this is Introduction or Discussion material

L1444 “Photos” instead of “Snapshots”

L146-153  No Results here; move to Intro/Methods if needed

Table 1 suggest transposing table (e.g., so Area data would be in a column instead of a row)

Table 2 suggest only capitalizing the first word in each column title,  and average height from ground data should all have same number of decimal places (I suggest fewer).

L162 long sentence, suggest divide into two

L170 “red tree crowns” unclear – is this western redcedar?

L170-180 This is introduction information.

L193 don’t capitalize “white pine”

L200 be consistent with use of common names or scientific names

Author Response

  1. The Introduction would benefit from addition of an objectives statement at the end.
  2. Response: objectives of the study were added at end of the introduction.
  3. The Methods section could be enhanced by adding description of the climate and forest composition and structure, and scientific names provided for all species common names only at the first instance.
  4. Response: two paragraphs of the climate and forest composition were added in the method section.
  5. Also, the Methods section needs to clearly explain if this study focused on damage to white pine (L90) or blue pine (L109) or counted damage on all species (L113).
  6. Response: now the sentence is clear which represented that this study is all about the Himalayan white pine.
  7. Usually, scientific names are italicized.
  8. Response: All the authors read the manuscript carefully and italicized all the scientific names in the manuscript.
  9. L60 add citations for black bears damaging faster-growing trees e.g.:
  10. Response: Citations have been provided at the end of the sentence.
  11. Dagley, C.M.; Berrill, J-P.; Leonard, L.P.; Kim, Y.G. 2018. Restoration thinning enhances growth and diversity in mixed redwood/Douglas-fir stands in northern California, USA. Restoration Ecology 26(6): 1170-1179.
  12. Response: Recommended citation has been provided at the end of the sentence
  13. Perry, D.W.; Breshears, L.W.; Gradillas, G.E.; Berrill, J-P. 2016. Thinning Intensity and Ease-of-Access Increase Probability of Bear Damage in a Young Coast Redwood Forest. Journal of Biodiversity Management and Forestry 5(3):1-7.
  14. Response: Recommended citation has been provided at the end of the sentence.
  15. L64 citation error – Nishi et al. should be numbered?  I believe you mean “severe”, but suggest rewording to “A single black bear damaged 60-70 trees during foraging in [forest type at location]”
  16. Response:

-Citation error has been resolved, proper number provided for Nishi et al.

-Rewording suggestion by the reviewer has been made according to the recommendations. Now the sentence is clear and meaningful.

  1. L65 check citation numbering as I do not think [10] Berrill et al. reported number of trees damaged by a single bear.
  2. Response: Proper and correct citation has been provided at the end of this sentence.
  3. L68 add objectives statement with research questions and/or hypotheses.
  4. Response: research objectives and research questions have been provided at the end of the introduction section.
  5. L78 Consider removing “GPS….” and only report the coordinates (not how they were collected).
  6. Response: reformatted the sentence according to the recommendations of the reviewer.
  7. L105 how was age of claw marks assessed? (detail please)
  8. Response: age of claw marks was calculated based on the nature of claw marks concerning the time, the entire damages were generally divided into different broad categories for easy assistance and data collection, data collection about the age of claw marks was also supported by field experts who already worked on this aspects.
  9. L107-108 this sentence not a method.
  10. Response: irrelevant sentence has been deleted from the method section.
  11. L112 I do not understand “measurable scale” - please clarify
  12. Response: the term measurable scale was replaced by measuring tape (10m), which was used for the length and width of each damage.
  13. L117 how were the transects located – random start point and azimuth? Systematic?
  14. Response: Line transect were randomly selected for the collection of data where a high number of damages were observed.
  15. L121 can ER also be reported as percent damaged? (i.e., were undamaged trees also tallied?) Reporting percent of all trees damaged is more intuitive/interpretable than number/transect length.
  16. Response: ER rate was determined by their standard as used in many research articles, formula for the ER is (ER: number of sign/transect length). It’s the frequency of specific signs that are repeated in a transect.
  17. L127 long sentence, and unclear if 0.4572 is a scar length or area or ?
  18. Response: Now the sentence is clear and meaningful, along with the size of damages represented in meter.
  19. L128 “…from the ground level…” is more of a method than a result
  20. Response: modified the sentence accordingly.
  21. L129-103 is blue pine the same species as white pine? I think this will be clearer to the reader once the Methods section is clarified per previous comment.
  22. Response: Yes blue pine and Himalayan White Pine are the same names for the Pinus wallichiana.  To increase the readability we changed to a single common name throughout the manuscript, that is “Himalayan White pine”.
  23. L134 please specify that the length and width refer to the scar (not the tree itself)
  24. Response: According to the reviewer's recommendation, the sentence was changed to scars.
  25. L130-138 Some of this is Methods mixed in with Results – please move Methods to the Methods section
  26. Response: methods sections were removed from this portion, now this paragraph is neat and clear which only represents the result section.
  27. L140-142 Not a Result – this is Introduction or Discussion material
  28. Response: L140-142 were moved to the discussion section where it best suited. Now this section is clear.
  29. L1444 “Photos” instead of “Snapshots”
  30. Response: the word “snapshots” is replaced by “Photos”.
  31. L146-153  No Results here; move to Intro/Methods if needed
  32. Response: L146-153 this paragraph has been moved to the method section, where it is best suited.
  33. Table 1 suggest transposing table (e.g., so Area data would be in a column instead of a row)
  34. Response: Modified according to the suggestion of reviewers.
  35. Table 2 suggest only capitalizing the first word in each column title,  and average height from ground data should all have same number of decimal places (I suggest fewer).
  36. Response: each word has been capitalized in each column of Table 2,  all the decimals in the average column are the same. Tabel two was modified according to the suggestion of the reviewer.
  37. L162 long sentence, suggest divide into two
  38. Response: the long sentence is divided into two sentences.
  39. L170 “red tree crowns” unclear – is this western redcedar?
  40. Response: The name of the tree is updated and modified to western redcedar.
  41. L170-180 This is introduction information.
  42. Response: The paragraph has been modified according to the reviewer's recommendation.
  43. L193 don’t capitalize “white pine”
  44. Response: Uncapitalized the word “White Pine”.
  45. L200 be consistent with use of common names or scientific names
  46. Response: scientific names were used for the mention sentence accordingly.
  47. The authors appear to have good understanding and use of the English language but I recommend editing by someone who regularly writes in English for clarity.
  48. Response: All the authors read and removed grammatical and technical language mistakes.
  49. Also, I recommend adding discussion of the limitations of the study and recommendations for future research, and possible implications for management/conservation/restoration.
  50. Research Limitations and future directions have been added at the end of the conclusion section.